# Sulfate-Reducing Bacteria of the Oral Cavity and Their Relation with Periodontitis—Recent Advances

**DOI:** 10.3390/jcm9082347

**Published:** 2020-07-23

**Authors:** Ivan Kushkevych, Martina Coufalová, Monika Vítězová, Simon K.-M. R. Rittmann

**Affiliations:** 1Department of Experimental Biology, Faculty of Science, Masaryk University, Kamenice 753/5, 62500 Brno, Czech Republic; 474593@mail.muni.cz (M.C.); vitezova@sci.muni.cz (M.V.); 2Department of Molecular Pharmacy, Faculty of Pharmacy, Masaryk University, Palackého tř. 1946/1, 61242 Brno, Czech Republic; 3Archaea Physiology & Biotechnology Group, Department of Functional and Evolutionary Ecology, Universität Wien, Althanstraße 14, 1090 Vienna, Austria

**Keywords:** sulfate-reducing bacteria, SRB, oral cavity, dental plaque, periodontitis, periodontal disease, sulfate, hydrogen sulfide, meta-analysis

## Abstract

The number of cases of oral cavity inflammation in the population has been recently increasing, with periodontitis being the most common disease. It is caused by a change in the microbial composition of the biofilm in the periodontal pockets. In this context, an increased incidence of sulfate-reducing bacteria (SRB) in the oral cavity has been found, which are a part of the common microbiome of the mouth. This work is devoted to the description of the diversity of SRB isolated from the oral cavity. It also deals with the general description of periodontitis in terms of manifestations and origin. It describes the ability of SRB to participate in its development, although their effect on periodontal inflammation is not fully understood. The production of hydrogen sulfide as a cytochrome oxidase inhibitor may play a role in the etiology. A meta-analysis was conducted based on studies of the occurrence of SRB in humans.

## 1. Introduction

Despite the improving oral hygiene of the human population and the availability of professional oral care, the number of inflammatory diseases of the oral cavity has been increasing recently [1]. One of the most common diseases is periodontitis, which is caused by a change in the microbial composition of the biofilm in the periodontal pockets [2]. In this context, an increased incidence of sulfate-reducing bacteria (SRB) has been found in the oral cavity, which are a common part of the human oral microbiome [3,4].

SRB are a heterogeneous group of naturally occurring microorganisms that share the ability of dissimilatory sulfate reduction to hydrogen sulfide (H_2_S) [5,6,7,8,9,10,11,12]. They occur in the external environment and as part of the microbiota of organisms, not only in the intestine but also in the oral cavity [3,4,10,11].

Periodontitis is a chronic inflammatory disease of the periodontal tissues with an uncertain and not well researched etiology [13,14]. However, its development is related to a change in the bacterial constitution of the dental plaque biofilm [15]. Clinical attachment loss is the predominant clinical manifestation and determinant of periodontal disease and its loss is a sign of destructive (physiologically irreversible) periodontal disease. Periodontitis is a disease with a variety of clinical manifestations, and the clinical description is continually being refined and updated [16].

In connection to the observed increase of incidence of SRB in the oral cavity in patients with periodontitis, the possibility of toxic effects of H_2_S on oral epithelial cells is being considered, which could lead to the onset and further development of the disease [5,6,7,17]. H_2_S acts as an inhibitor of cellular cytochrome oxidase, and may also have a secondary effect, by breaking down disulfide bonds in proteins, which affects granulocytes and their function within the immune system [18,19].

The aim of this study is to summarize the species diversity of SRB in the oral cavity, to characterize periodontitis and describe the connection between SRB and the products of their metabolism with the etiology of this disease. A meta-analysis of current studies on the occurrence of SRB in healthy humans and patients with periodontitis is also performed.

## 2. Basic Characteristic of Sulfate-Reducing Bacteria

### 2.1. Ecology and Diversity of Sulfate-Reducing Bacteria

SRB are a diverse group of prokaryotic organisms that share the ability of using sulfate (SO_4_^2−^) or other oxidized sulfur compounds as a terminal electron acceptor in the oxidation of organic matter [20,21,22,23,24]. Most SRB species described to date can be divided into the following phylogenetic groups based on rRNA sequence analysis: Gram-negative mesophilic SRB; Gram-positive spore forming SRB; Gram-negative thermophilic SRB; thermophilic archaeal SRB [25]; and a fifth group, the *Thermodesulfobiaceae* [26].

Representatives of SRB are found in almost all environments on Earth. Most strains are of environmental origin: they can be found in waters (freshwater, marine and brackish) and their sediments; soils; extreme conditions of geothermal environments; and hot springs, but can also be isolated from animals, including humans (sheep stomach, digestive tract of insects and mammals, including humans). Although SRB are anaerobic, in mixed populations (biofilm and microbial plaques) they can grow even in an originally aerobic environment. They tolerate temperatures from 5 to 75 °C and show considerable adaptability to new temperature conditions; can grow in anaerobic conditions or in water at a pressure of 1 × 10^5^ kPa; tolerate pH values in the range from 5 to 9.5; and also a wide range of osmotic pressure, but always only under the conditions of the presence of free sulfate [27,28,29].

Therefore, SRB are able to adapt to almost any natural environment on planet Earth, except for the usual aerobic environment. However, they can be isolated from almost any randomly taken soil or water sample, which means that they can survive long exposure to molecular oxygen (O_2_) and become active again when returned to anaerobic conditions [27].

### 2.2. SRB in Oral Cavity

SRB were first isolated from the oral cavity in 1995 [3]: genera *Desulfobacter* and *Desulfovibrio*. Further studies have shown their presence and described the diversity of other genera [4,30,31]. Therefore, we can state, that SRB isolated from the oral cavity are Gram-negative mesophilic bacteria belonging to the class *Deltaproteobacteria* of the phylum Proteobacteria. In this class we can find two orders from SRB: *Desulfovibrionales* and *Desulfobacterales* [32]. This section describes SRB that were isolated and described only in connection with the oral cavity, because the group of sulfate-reducing bacteria is very numerous in terms of species and most of the species are found only in the environment.

#### 2.2.1. Order *Desulfovibrionales*

The order *Desulfovibrionales* belongs to the phylum *Proteobacteria*, class *Deltaproteobacteria* and includes four families—*Desulfovibrionaceae*, *Desulfomicrobiaceae*, *Desulfohalobiaceae*, and *Desulfonatronumaceae* [33]. Bacteria belonging to the order Desulfovibrionales have cells in the shape of straight or slightly curved rods, which are often motile by one or more polar flagella. They are strictly anaerobic, and their growth is organotrophic or chemolitotrophic. The electron acceptor is usually sulfate, which is reduced to sulfide; some strains can reduce thiosulfate or elemental sulfur to sulfate or nitrates to ammonia. Simple organic substances serve as a carbon source and electron donor. Most species are mesophilic, only some are slightly thermophilic. They grow under neutral conditions, but some require an alkaline environment. Mesophilic species were isolated from aquatic environments (e.g., freshwater, marine and water supply) or from the digestive tract; several cases of isolation from clinical material have also been reported (but without pathogenic detection of isolates). Thermophilic species were found in the environment of geothermal marine springs [32,34].

##### Family *Desulfovibrionaceae*, Genus *Desulfovibrio*

To the family *Desulfovibrionaceae* also belong the genera *Lawsonia* and *Bilophila*, but they do not use sulphate as an electron acceptor and therefore cannot be designated as SRB. The genus *Desulfovibrio* is the most researched SRB genus—it can be relatively easily isolated and purified because, unlike most other SRB, bacteria of the genus *Desulfovibrio* are relatively resistant to oxidative stress and are therefore easier to cultivate [33]. They are mesophilic (optimal growth temperature 25–35 °C) and can also be halophilic [27].

Bacteria of the genus *Desulfovibrio* are Gram-negative non-sporulating and curved rod-shaped to spiral cells that require an anaerobic environment. Sulfate, often also sulfide or thiosulphate, is reduced to H_2_S; some substitutes may also reduce elemental sulfur. The electron donor is H_2_, lactate, ethanol and often also malate or fumarate; organic substances are not completely oxidized to acetate [32,34,35,36,37].

The genus contains more than thirty species. They are isolated from the O_2_-free aquatic environment, oil fields, industrial wastewater, the gastrointestinal tract of insects and vertebrates, excrements and the genital tract and the oral cavity of humans and mammals. They may be present asymptomatically in the gastrointestinal tract or may manifest as opportunistic pathogens and cause human infections (abdominal abscess, cholecystitis and other diseases) [35,36,37,38].

All representatives contain the sulphite reductase (desulfoviridine), which, together with their shape and active movement, safely distinguishes them from other genera [27,39]. The type species is *Desulfovibrio desulfuricans* [40]. It has the shape of a vibrio measuring 0.5 − 0.8 × 1.5 − 4.0 µm and is mobile.

*Desulfovibrio piger*, formerly *Desulfomonas pigra* [41], is rod-shaped measuring 0.8 − 1.3 × 1.2 − 5.0 µm. It occurs in the intestines of animals [33,42,43,44] and, as one of few species of the genus *Desulfovibrio*, is immobile (reflecting its species name, which means “lazy” in Latin).

*Desulfovibrio fairfieldensis*, together with the previously described genus *D. piger*, was isolated only in humans, not from the environment [41]. It is found in the subgingival biofilm, on the buccal mucosa and the dorsal surface of the tongue [45,46].

##### Family *Desulfomicrobiaceae*, Genus *Desulfomicrobium*

Bacteria of the genus *Desulfomicrobium* are non-sporulating gram-negative rod cells, usually motile with one or two polar flagella. The genus *Desulfomicrobium* is metabolically identical to the genus *Desulfovibrio*, but is characterized by a rod-like morphology and content bisulphate-reductase with desulforubidine instead of deulfoviridine. These are mostly mesophilic species, but one thermophilic species *Desulfomicrobium thermophilum* has been discovered in a hot spring in Colombia [47]. The main focus of their occurrence is mud and sediments, but a new species *Desulfomicrobium orale* has been isolated from patients with periodontitis from periodontal pockets [4].

*Desulfomicrobium orale* is a rod-shaped cell, size 0.7 × 2.0 − 3.0 µm, mobile with one polar flagellum: non-sporulating, obligately anaerobic and with an optimal growth temperature of 37 °C (temperature range 25–39 °C) [4].

#### 2.2.2. Order *Desulfobacterales*

This order belongs to the phylum *Proteobacteria*, class *Deltaproteobacteria*, and contains the families *Desulfobacteraceae* and *Desulfobulbaceae* [32]. Only the genus *Desulfobacter*, from the family *Desulfobacteriaceae*, has been isolated from the oral cavity and gastrointestinal tract of humans [3]; other representatives are commonly found in O_2_-free freshwater or brackish sediments oil fields. Representatives belonging to the order *Desulfobacterales* have different morphologies—from cocci to rod-shaped cells of various lengths, multicellular fibers or cells forming aggregates. Most are movable with one or two polar flagella. They are strictly anaerobic and have a respiratory type of metabolism. They use simple organic molecules as electron donors; some species can also use molecular hydrogen [32].

##### Family *Desulfobacteraceae*, Genus *Desulfobacter*

These are mesophilic organisms with a growth optimum of 28–32 °C, but growth can also occur at 10 °C [48]. Representatives of this genus are non-sporulating curved or straight gram-negative rod-shaped to ellipsoidal cells, with rounded ends of size 1 − 1.5 × 1.7 − 3 µm occurring individually or in pairs [25]. Cell motility may vary depending on the method of cultivation; motile forms with one polar flagellum [48] have been observed. They are chemoorganotrophic bacteria with a respiratory type of metabolism: strictly anaerobic. Acetate, which is completely oxidized to carbon dioxide, is used as the electron donor and carbon source.

Sulfate and other oxidized sulfur compounds serve as electron acceptors and are reduced to H_2_S. They require media containing a reductant (electron donor) and vitamins for growth, and often also require higher concentrations of NaCl and MgCl_2_. *Desulfobacter postgatei* is characterized by its oval shape. It was classified as an SRB of the oral cavity in 1995 [3].

## 3. Periodontitis

Periodontitis is a chronic multifactorial inflammatory disease associated with a dysbiotic plaque biofilm and is characterized by the gradual destruction of the tooth suspension. Primary features include loss of periodontal tissue support, which results in clinically significant loss of tooth fixation in the neck and root of the tooth and radiologically evaluable loss of alveolar bone, formation and presence of true periodontal pockets between the gums and tooth, and bleeding gums [49]. Periodontal diseases are the most common chronic inflammatory diseases of adults, which, with insufficient care, can lead to tooth loss.

A distinction is made between infections involving only the gums (gingivitis) and infections affecting the deeper supporting tissues of the teeth (periodontitis). Periodontitis is a chronic inflammatory disease affecting the tissues supporting the teeth, the so-called periodontium [50]. This is the tissue surrounding the teeth and fixing them in their position; it also serves as the first barrier against the entry of impurities, bacteria and other harmful elements. The periodontium is made up of gingiva (gum tissue), cementum (outer layer of the roots of teeth), alveolar bone (bony sockets into which the teeth are anchored) and periodontal ligaments. Periodontal pockets are formed; permanent inflammation and infection makes it impossible for them to heal spontaneously, as is the case on the surface of the body. The primary cause of their formation is the accumulation of dental plaque, which causes gingivitis.

### 3.1. Microbiota of the Oral Cavity

The oral cavity is a complex and heterogeneous microbial habitat. The saliva contains microbial nutrients, but it is not a particularly suitable growth medium, because nutrients are present in low concentrations. In addition, the saliva also contains substances of antibacterial nature (lysozyme, lactoperoxidase and others). However, despite the activity of antimicrobials, food residues and cellular waste can provide high concentrations of nutrients near surfaces, such as the gums and teeth, thus creating favorable conditions for extensive local microbial growth. The mouth contains the second most diverse microbial community in the body. It is estimated that more than 700 different species of bacteria can colonize adult mouths—in the oral cavity, they inhabit the hard surfaces of the teeth and soft tissues of the oral mucosa [51]. Each individual usually contains 150 or more different bacterial species.

The oral microbiota plays a major role in maintaining a normal physiological environment in the oral cavity. It includes both commensal and opportunistic pathogenic microorganisms that normally live in symbiosis with their host, but in dysbiosis, when the balance of the oral ecosystem is disturbed, a number of diseases can develop (tooth decay, periodontitis, etc.). In addition to bacteria, there are also archaea, viruses, protozoa and microscopic fungi. The mouth is not a homogeneous environment for microorganisms, but offers several different sites for microbial colonization [52], such as teeth, gingival sulcus, gums, tongue, cheeks, lips, and hard and soft palate. These sites form a highly heterogeneous ecological system and support the growth of very different microbial communities [53]. Their characteristics change over the course of an individual’s life.

Bacterial colonization begins immediately after birth, when, depending on the method of birth (whether natural or caesarean section), characteristic bacteria enter the oral cavity. Bacteria found in the oral cavity before eruption of the first teeth are mainly aerotolerant anaerobes, such as streptococci and lactobacilli, but other bacteria, including aerobes, occur in low numbers. After eruption of the first teeth, the balance shifts in favor of anaerobic species that are adapted to grow on the hard surfaces of the teeth. Colonization of the tooth surface begins with the attachment of individual bacterial cells. Excessive proliferation of these cells leads to the formation of a thick bacterial layer called dental plaque [52].

### 3.2. Dental Plaque

Dental plaque is a structurally and functionally organized biofilm firmly, adhering to surfaces in the oral cavity. It should be noted that this is not ordinary adhesion, but it is a selective adhesion thanks to the recognition system on bacterial surfaces. The biofilm is composed of hundreds of bacterial species, polymers and extracellular products of microbes. The microorganisms (microbiota) that make up the human microbiome are not just single-celled organisms living side by side, but form highly regulated, structurally and functionally organized communities attached to surfaces as biofilms [54]. Communication between oral microorganisms is essential for initial colonization and subsequent biofilm formation on dental enamel.

Dental plaque can be divided into supragingival and subgingival. The predominant organisms of supragingival plaque are Gram-positive facultatively anaerobic bacteria, especially the species *Actinomyces* sp., *Streptococcus* sp. and *Capnocytophaga* sp. Gram-negative species, including *Veillonella* sp., *Prevotella* sp., *Porphyromonas gingivalis* and *Tannerella forsythia*. Once the supragingival plaque is formed and the ecosystem balance is established, a subgingival plaque begins to form. Its composition has been studied more thoroughly because subgingival deposits are involved in periodontal diseases. Subgingival plaque includes the following species: *Streptococcus* sp., *Prevotella denticola*, *Porphyromonas endodontalis* and *Porphyromonas gingivalis* [55].

In 1994, Haffajee and Socransky summarized criteria that helped define subgingival bacterial species as periodontal pathogens [56]. These criteria include:the relationship of the species to the disease;possibility of elimination of the bacterial organism by treatment;destructive host response;production of virulence factors and mechanism of pathogenicity;studies of pathogenicity in animal models;risk assessment.

#### Microbial Complexes of Dental Plaque

Studying microbial complexes in dental plaque helped to clarify the interrelationships between bacteria and their communities in the biofilm. It can be assumed that different proportions of individual complexes—and thus, of opportunistic pathogens—cause a difference between healthy and diseased oral tissue.

Socransky et al. [57] examined over 13,000 samples of subgingival plaque from 185 test subjects. By cluster analysis, they identified 32 taxa which formed into six specific groups of microorganisms (Table 1). These groups have been marked with colors for easier orientation. Bacteria of the purple, yellow and green complex correspond to periodontal health; bacteria of the predominantly red complex, but also the orange complex, together with other unclassified species, are likely periodontal pathogens.

The study was followed up, in 2008, with the aim of similarly simplifying the classification of supragingival plaque taxa [58]. The test sample for cluster analysis consisted of 187 individuals and a total of almost 5000 supragingival plaque samples. A total of 40 taxa were identified which, after analysis, formed six phylogenetically distinct groups (Table 2). The research results, to some extent, reflect previous research on subgingival plaque.

### 3.3. Etiology of Periodontitis

The study of the etiology of periodontitis has undergone a long development, and yet many questions remain unclear. It is generally accepted that both internal and external influences must be present for the onset of periodontitis. Oral biofilms, called dental plaque, were first observed by A. van Leeuwenhoek in the seventeenth century [59] and are associated with the most common oral diseases: tooth decay and periodontal disease. To effectively treat and prevent these diseases, it is important to understand how healthy dental plaque can develop into a pathological condition. Perceptions of the relationship between changes in dental plaque and the shift from oral health to disease have changed over time [59].

#### 3.3.1. Non-specific Plaque Hypothesis

Non-specific plaque hypothesis (NSPH) is a theory that assumes that pathogenicity is determined by the amount of plaque without distinguishing between individual types of bacteria or their virulence—the disease is caused by a non-specific increase of bacteria [50,60].

This theory dates to the nineteenth century, when Willoughy Miller [61], a student of Robert Koch, wanted to identify the bacterial species responsible for caries. Due to limited taxonomic data on oral bacterial agents and ignorance of the various sites of their occurrence within the oral cavity, he concluded that caries is bacteriologically non-specific. Miller and others have argued this: acid demineralizes the tooth, and all plaque bacteria produce acid, so all bacteria contribute to tooth decay, especially when they accumulate in places that are difficult to keep clean [61].

If this were true, the host would have a threshold capacity to detoxify bacterial products (e.g., saliva that neutralizes acid) and the disease would develop only if this threshold was exceeded and virulence factors could no longer be neutralized [62]. The conclusion is that if any plaque has the same potential to cause disease, the best way to prevent disease would be to mechanically remove as much plaque as possible, e.g., with a toothbrush or toothpick. These measures have not been very effective in preventing tooth decay, but cleaning the teeth and interdental spaces with a toothbrush and floss has reduced the risk of gingivitis and has become the main treatment method for preventing periodontal disease [62]. Improvements in bacterial isolation and identification techniques in the mid-20th century led to the abandonment of NSPH.

#### 3.3.2. Specific Plaque Hypothesis

The specific plaque hypothesis (SPH) suggests that of all the different organisms that make up the oral microbiota, only a few species are actively involved in the onset of the disease. Researchers noticed an obvious lack of correlation between the amount of accumulated plaque and the destruction of periodontal tissues. Keyes [63] wrote: “Although it is rare, most dentists at some time in their careers have seen patients with so-called “dirty mouths,” yet with negligible if any caries or periodontal disturbances. One possibility is that such mouths may not harbour odontopathic microorganisms.” SPH was also supported by the finding that there are differences in the composition of dental plaque in periodontally healthy and diseased sites and individuals.

The theory of specific microorganisms associated with various periodontal diseases has become a generally accepted dogma. Therefore, several scientific groups set out to search for responsible oral pathogens for various periodontal conditions [63].

In 1976, Walter J. Loesche proposed a hypothesis, assuming that tooth decay was caused by infection with specific bacteria in dental plaque [64]. In this hypothesis, he proposed the use of antibiotics directed against specific bacterial species as a treatment and prevention of tooth decay [64].

#### 3.3.3. Ecological Plaque Hypothesis

In 1994, Philip D. Marsh proposed a hypothesis that combined the key elements of the previous two hypotheses [65]. In his Ecological plaque hypothesis (EPH), he states that the disease is the result of an imbalance throughout the microbiome caused by ecological stress, which results in the enrichment of some “oral pathogens” or disease-related microorganisms (Figure 1).

Important features of EPH are that the selection of potentially pathogenic bacteria is directly associated with these changes in the environment, and moreover, the disease may not have a specific etiology; any species with relevant properties can contribute to the course of the disease. The prevention of the disease according to this hypothesis is the direct targeting of domestic periodontal pathogens (for example by the administration of antibiotics), together with the adjustment of the environment which is responsible for their enrichment [66].

#### 3.3.4. Keystone Pathogen Hypothesis

The term key species is derived from basic ecological studies. A key species is a species that plays a key role in the ecosystem and whose extinction (or even a significant reduction in its numbers) can disrupt the existing ecosystem or habitat. The role that a key species plays in its ecosystem is analogous to the role of the central stone of the arch in architecture—even if it is subjected to the least pressure of all the stones of the arch, without it the arch would collapse. Thus, the key species has a disproportionate impact on the environment due to its overall abundance [67].

The Keystone Pathogen Hypothesis (KPH) for (oral) microbiology was introduced by George Hajishengallis and colleagues [67]. They reported that certain low-incidence microbial pathogens can cause inflammatory diseases by increasing the number of common microorganisms and changing their species composition. Unlike dominant species, which can affect inflammation by their abundance, key pathogens can cause it, even though they are present in low numbers.

As the disease progresses to advanced stages, the key pathogen is detected in a greater number [57]. Importantly, although their absolute number increases, key pathogens may reach a relatively lower number compared to the total number of bacteria that increases as with dental plaque accumulates [68].

#### 3.3.5. Summary of the Hypothesis

Periodontal diseases, gingivitis and periodontitis cannot be explained by just one of the hypotheses (comparison in Table 3). The intimate interaction of bacteria and hosts leading to an inflammatory response increases the complexity of these diseases.

Periodontitis is the result of complex interactions between microorganisms and the host immune system [69]. Due to the increased amount of plaque, virulent bacteria and key pathogens themselves, the concentration of inflammatory response mediators increases [70,71]. Increased concentrations of anti-inflammatory cytokines in the periodontium can directly affect bone loss [72].

However, it is also necessary to take into account that there are differences in susceptibility to oral disease among people, even if they share the same lifestyle. There are people who have suffered from gingivitis all their lives, but never develop periodontitis, while others show a transition to periodontitis after a short episode of gingivitis. This can be caused by different ratios of bacterial species, which is determined, among other things, by genetic factors.

None of the currently available hypotheses can satisfactorily combine the actual behavior of microorganisms and the host which leads to the maintenance of health or its shift to disease. Therefore, further research is needed, which will hopefully lead to the discovery of the mechanisms that determine the transformation of healthy teeth into unhealthy ones [72].

### 3.4. Clinical Picture of Periodontitis

Periodontal diseases are manifested by several elementary pathological changes. Along with gingival inflammation, the basic clinical manifestations of periodontitis are, most commonly, periodontal pockets, related to the loss of alveolar bone caused by its inflammatory resorption. The periodontal pocket is the free space between the root of the tooth and the gums, which arises after the lowering of the alveolus and after the displacement of the epithelial attachment to the tooth (see Figure 2). 

The true periodontal pocket is caused by the loss of tooth attachment; it is deeper than 3.5 mm. Other symptoms of the disease are changes in the position of the teeth and their increased mobility, the formation of pus in the periodontal pocket and the emergence of painful complications (e.g., retrograde pulpitis and periodontal abscess).

### 3.5. Classification of Clinical Forms of Periodontitis

The classification scheme for periodontal disease is essential for clinicians and scientists. Disease classification systems allow clinicians to create a structure that is used to identify diseases in relation to etiology, pathogenesis and treatment. This makes it possible to better organize and streamline the patient’s treatment. It provides researchers with the opportunity to better investigate the etiology of the disease, pathogenesis, observe natural development and suggest treatment. Classification systems also allow clinicians and scientists to communicate in a common language based on defined systematic names. However, the pathobiological nomenclature is not precisely and definitively given. The recognition and treatment of periodontal disease can be traced back to antiquity, and many classification theories have emerged since then. Periodontitis is not static and can develop during the disease, so its classification is constantly evolving and responds to recent scientific and clinical findings.

#### 3.5.1. Classification of the American Academy of Periodontology 1999

This classification scheme was developed by the American Academy of Periodontology (AAP) in 1999 [73]. It was developed encyclopedically; it was very complete, detailed, and comprehensive, but in its entirety, it was not entirely suitable for routine use by experts. A simplified and more satisfactory summary is shown in Table 4.

Although the 1999 AAP classification of periodontitis no longer applies (as it was replaced by a new resolution from 2017), it is mentioned in this work for the following reasons: the vast majority of articles from which the work draws are based on sources older than 2017; the classification from 2017 is relatively new, so it is appropriate to mention the previous classification for completeness.

Chronic periodontitis is a form of destructive periodontal disease that is generally characterized by slow progression. Aggressive periodontitis is a diverse group of highly destructive forms of periodontitis affecting primarily young individuals, including conditions previously classified as “early-onset periodontitis” and “rapidly progressing periodontitis”. Periodontitis as a manifestation of a systemic disease is a heterogeneous group of systemic pathological conditions that include periodontitis only as a manifestation. Necrotizing periodontal disease is a group of conditions that share a characteristic phenotype in which necrosis of the gums or periodontal tissues is a prominent manifestation. Periodontal abscesses represent a clinical entity with different diagnostic features and treatment requirements [49].

#### 3.5.2. Classification of the American Academy of Periodontology 2017

Groups of top experts in periodontology and implantology worked on the new classification, and their proposals were approved by the AAP and the European Federation of Periodontology (EFP) [75]. The current revised classification system recognizes, on the basis of pathophysiology, three different and clearly distinguished forms of periodontitis [49]: (A) necrotizing periodontitis, (B) periodontitis as a direct manifestation of systemic disease and (C) periodontitis (Table 5).

The differential diagnosis of necrotizing periodontitis is based on the history of the disease and specific symptoms, or on the presence or absence of an unusual systemic disease that alters (weakens) the hosts’ immune response. Typical features for this group are necrosis of the interdental papillae, bleeding and soreness. Classification of periodontal disease as a direct manifestation of systemic disease should be based on the primary disease. It includes some rare conditions that directly affect the health of the periodontium.

The remaining clinical cases of periodontitis are diagnosed as periodontitis. Periodontitis is diagnosed in a patient if: 1) there is a clinical attachment loss in the interdental space in two or more non-adjacent teeth, or 2) there is a buccal or oral clinical attachment loss ≥3 mm together with a periodontal pocket >3 mm for two or more teeth. For more accurate descriptions of periodontitis, staging and grading are used, similarly to other medical disciplines. Staging depends on the severity of the disability (due to the destruction of the periodontium proportional to the length of the root and also the loss of teeth due to the disease). Another factor is the complexity of the disease (depth of periodontal pockets, the presence of defects, the degree of tooth mobility and more). Accordingly, it includes four stages (I to IV). After treatment, despite the elimination of factors, staging remains the same.

Grading provides additional information on the biological properties of the disease, including an analysis of its rate of progression. It considers risk factors that may affect the further development and course of the disease and assesses whether the disease or its treatment may have an adverse effect on the overall health of the patient. Accordingly, they are divided into three groups: slow, medium and fast progression. Grading, unlike staging, is not constant and the rate of progression may change during treatment. A summary and illustration of the distribution is shown in Table 5.

### 3.6. Epidemiology and Treatment

In periodontal diseases, it is particularly difficult to detect the initial stage of the disease, when changes in the shape and color of the papillae are not very pronounced and frequent symptoms of changes in the suspension apparatus cannot be clinically recognized at all. Therefore, the data obtained to determine the incidence of the disease may be distorted. The incidence of periodontal diseases is generally considered to be very high. The disease occurs in all age groups. In addition, periodontitis is now the sixth most common oral disease in the world [77] and affects between 5% and 20% of all adults [78].

Periodontitis is a curable disease. The basis is the introduction of proper oral hygiene and its long-term maintenance, removal of subgingival plaque and treatment of periodontal pockets leading to their healing or removal. This can be achieved in two ways: conservative therapy and surgical treatment. At present, both therapeutic approaches are usually combined and physicians often also encourage the formation of supportive tissues (targeted formation of dental cementum, alveolar bone and periodontal fibers). To maintain the result, long-term care is necessary after the inflammation has been eliminated and the tissues have healed [79].

## 4. SRB and oral tissues

The health or disease of oral tissues depends on the interaction between the host and the microbial community. As described, the etiology of periodontitis is likely to be multifactorial and includes both environmental, genetic and immune factors, but is particularly affected by the activity of oral microbiota. Relatively recently, the occurrence of SRB was found in places of the oral cavity that showed clinical signs of periodontitis [3,4,30,31,45]. The relationship between SRB and periodontitis has long gone unnoticed, mainly due to their slow growth and the possibility of overgrowth by other species in mixed cultures [80,81,82].

SRB occur in the mouths of approximately 10% of healthy people. In patients with periodontitis, the frequency of their occurrence is significantly higher, reaching up to 86% [45], and is associated with increased depth of periodontal pockets and their bleeding [82]. These facts suggest that SRB are part of normal oral microbiota, but prefer periodontal pockets, which are ideal for the growth of these bacteria given their anaerobic growth conditions.

### 4.1. Sources of Sulphate in the Oral Cavity

The amount of SRB in the oral cavity is limited by the amount of sulfate available. Potential sources of sulfate in the subgingival region include free sulfate in the pocket fluid and glycosaminoglycans and sulfur-containing amino acids (cysteine and methionine) from periodontal tissues [80,81,82,83,84,85]. SRB then metabolize the sulfate to H_2_S [86,87,88,89,90,91].

The amount of H_2_Sis also affected by eating habits. It is found in many food products (beer, cheese, wine, bread, canned meat and vegetables, pickled products) where it serves as a preservative, antioxidant or bleaching agent [92]. Organic polysulfides naturally found in garlic, onions, *Brassicaceae* plants (e.g., cabbage, cauliflower, cabbage, broccoli), and durian can directly release H_2_S through interactions with protein thiols or intracellular thiols (e.g., glutathione) [93]. Furthermore, a significant amount of substrate for SRB can be released from mucin, e.g., by sulfatases, which can be produced by several species of oral streptococci [94].

### 4.2. The Role of SRB as an Etiological Agent

Proteolytic activity in the mouth is an important factor in the development of bad breath [95,96]. Volatile sulfur compounds (VSCs) are gases that are primarily responsible for halitosis, a condition in which undesirable unpleasant odors are present in the mouth. VSCs are derived from bacterial protein metabolism. These metabolites include many compounds, such as skatole, H_2_S, methanethiol and dimethyl sulfide. Studies have shown that these compounds are toxic even at low concentrations. In other words, VSCs may also contribute to the etiology of both gingivitis and periodontitis [80].

SRB are a heterogeneous group of naturally occurring species that share the ability of dissimilatory sulfate reduction to H_2_S. H_2_S is a highly toxic agent (which causes cell damage in a manner similar to cyanide) with lethal effects due to inactivation of cytochrome oxidase [97]. Cytochrome oxidase is a transmembrane protein of mitochondria, the last enzyme in the respiratory electron transport chain. O_2_ cannot bind to it after its blocking, which results in the impossibility of ATP synthesis and thus energy production. The sulfide was detected in deeper periodontal pockets, at concentrations high enough to inactivate cellular cytochrome oxidase [98]. Accumulation of H_2_S can also have a secondary toxic effect—it has the ability to cleave disulfide bonds in proteins, react with them and bind various metal ions [99]. This may affect the ability of granulocytes to opsonize microorganisms, leading to suppression of the immune response in the periodontal pocket [100]. H_2_S also inhibits myeloperoxidase and catalase [101].

Thus, the increase in SRB in the pockets depends on certain tissue-degrading microbial activities. In addition, bacteria produce the virulence factor H_2_S, which may enhance the process of periodontal tissue destruction, so the presence of SRB could serve as an indicator of tissue degradation in the periodontal pocket [81].

## 5. Occurrence of SRB in the Population

This part of the work consists of an overview of literature data [45,81,102] that was used in creating a meta-analysis to compare the prevalence of SRB in healthy individuals and periodontitis patients. The meta-analysis was performed using Review Manager 5.3. Furthermore, a comparison of studies monitoring the occurrence of SRB in the population is presented, which could not be used to create a meta-analysis, but the results of which are satisfactory to illustrate a certain trend of the occurrence of these bacteria in the population.

The result of the meta-analysis is arranged as a graphical output, the so-called forest plot. It shows an estimate of the total effect as an aggregate of effects of each study included. The graph shows an estimated effect of each study and its weight, as well as confidence intervals. Studies are usually marked with the author’s name and the year of publication. The graph includes a contingency table of values with which the statistical software works. In the graph, the individual estimated magnitude of the effects is represented by the center of the square, whose size reflects the weight of the study. The horizontal lines passing through the square show the size and position of the confidence intervals (CI) of the individual estimates at the 95% confidence level.

A vertical line (y-axis) separates the positive effect from the negative one, thus indicating a zero effect. A logarithmic scale is marked on the horizontal axis (x-axis), which allows a symmetrical display of confidence intervals. If a square or confidence interval intersects the zero-effect axis, the study data are statistically insignificant. If the axis is not intersected, it means that the study results are statistically significant at the 95% confidence level.

The total magnitude of the effect is shown in the graph by a diamond, whose center indicates the weighted effect size of the individual studies. Its width is given by the confidence interval for the calculated total effect. The forest plot is able to show to what extent the data from several studies, observing the same effects, overlap. Results that do not overlap well are called heterogeneous and such data are less conclusive. Heterogeneity (I^2^), given as a percentage, represents the overall measure of variability in the effect of the studies that is due to heterogeneity. These values are often classified into three categories: none/low (<25%), medium (25–50%) and high (>50%) heterogeneity.

The incidence of SRB was examined in a group of healthy people and patients with periodontitis (Figure 3). The location of the square on the right indicates the value of the absolute risk ratio (probability of occurrence). The absolute risk ratio is calculated as the probability of the occurrence of SRB positive results (persons) in a group of patients with periodontitis divided by the probability of the same phenomenon in healthy persons. It follows that the occurrence of SRB is less common in healthy individuals.

All three studies show a significant deviation (none intersects the zero-effect axis, so the results are statistically significant at the 95% level of significance). The result of the I^2^ heterogeneity test is 11%, indicating consistency of results. The diamond is located on the right side of the axis. A summary of the studies found that SRB are less common in healthy individuals than in people with periodontitis, i.e., SRB have a certain association with the occurrence of periodontitis.

As mentioned above, the meta-analysis examines and compares studies where two samples of the population have been studied: healthy and periodontal-affected individuals. However, several studies have also been performed in the past that have observed only one of the groups and therefore cannot be included in the meta-analysis. In this chapter, they are (along with the studies already included) shown in a graph for visual comparison (Figure 4).

The first three studies in the graph are studies that have already been evaluated by a meta-analysis. We see a clear difference in the percentage of people positive for SRB between healthy and affected people, which are confirmed by our meta-analysis. The next four bars represent studies on periodontal patients [3,45,81,103]. The latest of these deviates somewhat from the percentage (it is low), which may be due to the small number of individuals in the sample. The last bar is devoted to a study in healthy patients, but with a different methodology [46]: patients were left without oral hygiene for 24 h before sampling, so the ratio does not correspond to previous studies, but may prove that SRB are part of normal oral microbiota.

## 6. Conclusions

SRB are a group of anaerobic microorganisms that share in common a reaction to obtain energy-dissimilatory sulfate reduction. These bacteria occur not only in the external anoxic environment, where they colonize water sediments and soil, but also in the body of humans and animals. In animals and humans, they colonize the intestine and the oral cavity, mostly periodontal pockets. To date, the genera *Desulfovibrio* and *Desulfomicrobium* and *Desulfobacter* have been isolated from the oral cavity.

One of the most common inflammatory diseases of the oral cavity is periodontitis, the abundance in the population is increasing. It is a disease with a vague mechanism of origin. It occurs in people with excess dental plaque, which supports the theory that microorganisms and their metabolites play a role in the development of this disease. The oral microbiome was divided into several groups according to taxonomic congeniality, with some complexes being more associated with periodontal disease, leading to the conclusion that some species of microorganisms have greater pathogenetic potential than others. 

Periodontitis is, to a certain extent, a treatable disease, the treatment of which combines conservative therapy with surgical treatment, the proper effect of which requires long-term oral hygiene care. The final product of the dissimilatory sulfate reduction is H_2_S, which has a toxic effect on oral epithelial cells. It acts as an inhibitor of cellular cytochrome oxidase and may also have a secondary effect by breaking down disulfide bonds of proteins, affecting granulocytes and their function within the immune system.

Our meta-analysis combines the benefits of clinical trials and epidemiological investigations. It relates the results of scientific studies examining a smaller number of individuals to other studies, thus creating a much larger sample and being able to better discover and describe the overall result. A meta-analysis of three studies looking at the presence of SRB-positive people in the oral cavity showed that the percentage of SRB-positive people is higher in people with periodontitis.

There are still many unanswered questions about the exact mechanism of periodontitis onset and whether it is really related to sulfate-reducing and/or other bacteria. It is therefore necessary to continue research into microbial populations of the oral cavity and their effect on the tissues they inhabit. Despite the fact that the article contains information on the importance of proper oral hygiene, it should be emphasized that, as most publications indicate, this simple method is only dependent on an individual understanding of the situation, and has a huge impact on remission and inhibiting the progression of the disease.

## Figures and Tables

**Figure 1 jcm-09-02347-f001:**
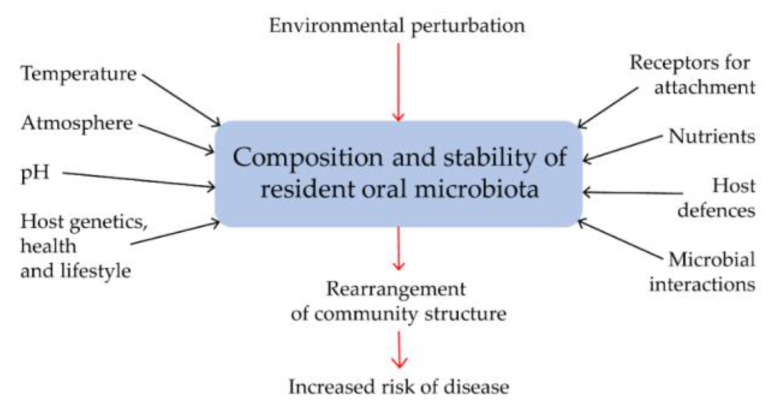
Host factors that affect the microbial composition, activity and stability of the oral microbiota [66]. A change in a key environmental factor can disrupt the natural balance of a resident microbiota (microbial homeostasis) and can lead to a rearrangement of the structure and activity of the microorganism community; such a change may make the site in the oral cavity prone to disease outbreaks.

**Figure 2 jcm-09-02347-f002:**
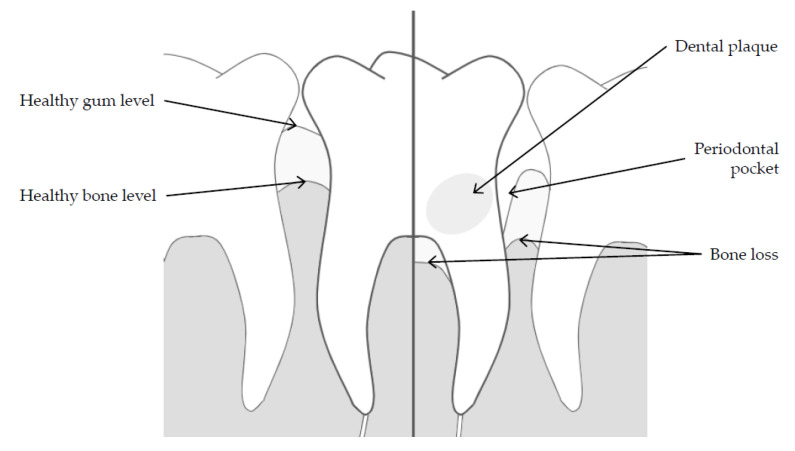
Schematic representation of the suspension apparatus of a human tooth, illustrating periodontal pocket. In the left part of the picture there is a healthy tooth; in the right, a tooth with periodontitis.

**Figure 3 jcm-09-02347-f003:**
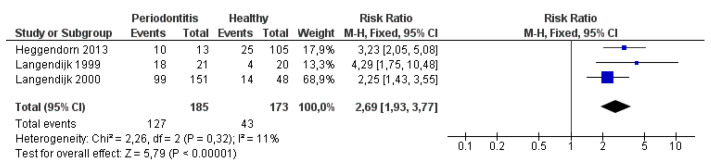
Meta-analysis of occurrence of sulfate-reducing bacteria (SRB) in a group of healthy people and patients with periodontitis.

**Figure 4 jcm-09-02347-f004:**
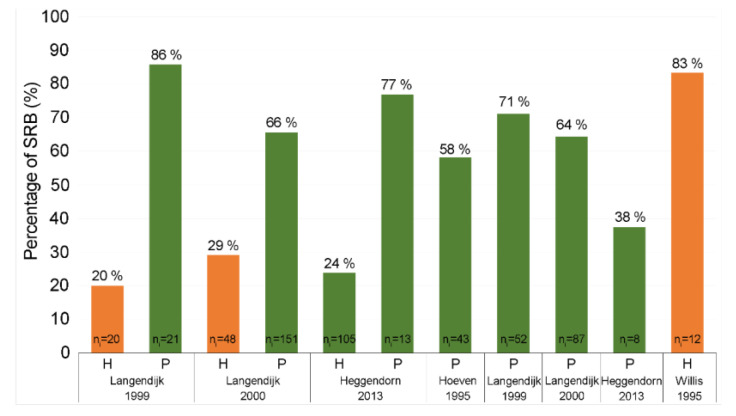
Comparison of the percentage of SRB in healthy people and patients with periodontitis. (H—healthy subjects; P—periodontitis; n_i_—number of individuals in sample).

**Table 1 jcm-09-02347-t001:** Classification of subgingival bacterial complexes [57].

Bacterial Species	Complex
*Actinomyces odontolyticus, Veillonella parvula*	Purple
*Streptococcus gordonii, S. intermedius, S. mittis, S. oralis, S. sanguis*	Yellow
*Capnocytophaga sputigena, C. gingivalis, C. ochracea, C. gracilis, Eikenella corrodens*	Green
*Campylobacter rectus, Fusobacterium nucleatum, Peptostreptococcus micros, Prevotella intermedia*	Orange
*Tannerella forsythia, Porphyromonas gingivalis, Treponema denticola*	Red
*Aggregatibacter actinomycetemcomitans, Selenomonas* sp.	Not clustered

**Table 2 jcm-09-02347-t002:** Classification of supragingival bacterial complexes [58].

Bacterial Species	Complex
*Veillonella parvula, Neisseria mucosa*	Purple
*Capnocytophaga sputigena, C. gingivalis, Eikenella corrodens*	Green
*Actinomyces odontolyticus, A. israelii, A. gerencseriae, A. naeslundii*	Blue
*Peptostreptococcus micros, Eubacterium saburreum, Streptococcus anginosus,* *S. constellatus, S. intermedius, S. mitis, S. oralis, S. sanguinis, S. gordonii*	Yellow
*Fusobacterium nucleatum, E. periodonticum, Campylobacter showae, C. rectus,* *C. gracilis, C. ochracea, Prevotella intermedia, P. nigrescens, P. melaninogenica*	Orange
*Treponema denticola, Eubacterium nodatum, Porphyromonas gingivalis, Tannerella forsythia*	Red

**Table 3 jcm-09-02347-t003:** Comparison of the different hypotheses [27].

Hypothesis	References	Bacteria Involved in Disease	Relates to Factors *	
			Ecological Changes	Host Specific
NSPH	Miller, 1890 [61]	all	-	-
	Theilade, 1986 [62]	all, difference in virulence	+	-
SPH	Loesche, 1976 [64]	specific bacteria	-	-
EPH	Marsh, 1994 [65]	all, enrichment of specific pathogenic bacteria	+++	-
KPH	Hajishengallis et al., 2012 [67]	specific bacteria, dependent on (some of) remaining microbiota	++	+

* Factors that could differ amongst hosts, e.g., innate immune system (levels of cytokine and toll-like receptors (TLR) expression), response to certain bacteria, Greatest Common Factor (GCF) properties (iron concentration), saliva properties (buffer capacity) and enamel repair. - not or only briefly mentioned, + mentioned, ++ mentioned and described, +++ described in detail.

**Table 4 jcm-09-02347-t004:** Summary of the classification scheme AAP 1999 [74].

1. Gingival diseases	(A) Plaque-induced
	(B) Non-plaque induced
2. Chronic periodontitis	(A) Localised
	(B) Generalised
3. Aggressive periodontitis	(A) Localised
	(B) Generalised
4. Periodontitis as a manifestation of systemic disease	
5. Necrotizing periodontal diseases	
6. Abscesses of the periodontium	
7. Periodontitis associated with endodontic lesions	
8. Developmental or acquired deformities and conditions	

**Table 5 jcm-09-02347-t005:** Summary of the classification scheme AAP 2017 [76].

1. Necrotizing Periodontal Diseases
(A) Necrotizing gingivitis
(B) Necrotizing periodontitis
(C) Necrotizing Stomatitis
2. Periodontitis as manifestation of systemic diseases
3. Periodontitis
(A) Stage: based on severity and complexity of management
Stage I.: initial periodontitis
Stage II.: moderate periodontitis
Stage III.: severe periodontitis with potential for additional tooth loss
Stage IV.: severe periodontitis with potential for loss of the dentition
(B) Extent and distribution:
localized; generalized; molar-incisor distribution
(C) Grade: evidence or risk of rapid progression, anticipated treatment response
Grade A: slow rate of progression
Grade C: rapid rate of progression

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
