# Peer review of "Sulfate-Reducing Bacteria of the Oral Cavity and Their Relation with Periodontitis—Recent Advances"

_jcm, 2020, doi:10.3390/jcm9082347_

Round 1

Reviewer 1 Report

Very interesting review paper which is an attempt to draw attention to the possible participation of sulfate-reducing bacteria as one of the etiological factors of periodontal disease - a multi-cause disease and recognized as a social one.  

All issues related to the problem of periodontal changes are presented in a transparent way: starting from the time-changing view on the share of dental plaque as one of the main etiological factors of periodontal disease, through historical earlier classifications of periodontal disease in relation to the new one valid from 2017.

The Authors interestingly combined the latest data on sulfide-reducing bacteria and their possible participation in the development of periodontal disease with basic information about this disease.

In addition, a meta-analysis of the occurrence of sulfate-reducing bacteria in healthy people and with periodontal disease allows estimating the possible share of these bacteria in the etiology of the disease.

I would have only a few suggestions that the Authors may or may not use. 

1. Addressing the symptoms of periodontitis would be advisable to add that clinical attachment loss (CAL) is the predominant clinical manifestation and determinant of periodontal disease and its loss is a sign of destructive (physiologically irreversible) periodontal disease.

  1. line 220 – when writing about the adhesion of bacteria to the surfaces of the teeth and oral mucosa, it would be worth noting that this is not ordinary adhesion and add one or two sentences about selective adhesion thanks to the recognition system on bacterial surfaces.
  2. Despite the fact that the article contains information on the importance of proper oral hygiene, it should be emphasized that, as most publications indicate, this simple method, unfortunately only dependent on an individual understanding of the situation, has a huge impact on remission and inhibiting the progression of the disease.

Author Response

Dear Reviewer,

we would like to thank you for reviewing our manuscript and your time.

Our manuscript was carefully revised and we are thankful for your important and critical comments, which improved our manuscript.

We have corrected our manuscript according to your important comments and recommendations.

We have also written our responses to your comments below:

I would have only a few suggestions that the Authors may or may not use.

1. Addressing the symptoms of periodontitis would be advisable to add that clinical attachment loss (CAL) is the predominant clinical manifestation and determinant of periodontal disease and its loss is a sign of destructive (physiologically irreversible) periodontal disease.

It was added. Thank you!

line 220 – when writing about the adhesion of bacteria to the surfaces of the teeth and oral mucosa, it would be worth noting that this is not ordinary adhesion and add one or two sentences about selective adhesion thanks to the recognition system on bacterial surfaces.

It was added.

Despite the fact that the article contains information on the importance of proper oral hygiene, it should be emphasized that, as most publications indicate, this simple method, unfortunately only dependent on an individual understanding of the situation, has a huge impact on remission and inhibiting the progression of the disease.

It was corrected.

Thank you for your time and your help again.

Best Regards,

The Authors

Reviewer 2 Report

Overall this is a well written paper that I think is worthy of publishing. However, I take a bit of issue with the title of the paper- this isn't really a meta-analysis, it is more of a review. The review section is fantastic but the meta-analysis is quite lacking. I think the paper should be titled and framed more as a review with the meta-analysis being such a small piece. Overall, I would accept this work with minor corrections rearranging and clarifying some of the content.

Abstract: I think it would be worthwhile to mention an example of SRB. The second to last sentence doesn't really seem to make sense. That point is barely mentioned in the article.

Body text: 

Line 112: Extra 'o'

Line 236-241: This seems out of place, perhaps make this a small chart or just a sentence? Why bullet points?

Tables 1 and 2 look nice. 

Would it be possible to move 3.4 to earlier in the text? Same with figure 4, it might benefit the text to put it earlier and include documentation of where those different plaques come from on the diagram. 

Table 3 has formatting issues.

Pg 10 mentions antibiotics, but this is not discussed in the text in detail. How do antibiotics impact the presence of these microbes? I'd like to see more on that. 

The meta-analysis is really only at the very end of the article and only deals with a few studies. I take it because so little research has been done? The explanation in 5.2 of the graphics (which I think should be a caption) is not quite clear. Are these limitations to these conclusions? How many studies of those examined actually look at SRB? I'm not totally convinced that a forest plot is the best way to address these data, maybe the other reviewers have suggestions

Author Response

Dear Reviewer,

we would like to thank you for reviewing our manuscript and your time.

Our manuscript was carefully revised and we are thankful for your important and critical comments, which improved our manuscript.

We have corrected our manuscript according to your important comments and recommendations.

We have also written our responses to your comments below:

Overall this is a well written paper that I think is worthy of publishing. However, I take a bit of issue with the title of the paper- this isn't really a meta-analysis, it is more of a review. The review section is fantastic but the meta-analysis is quite lacking. I think the paper should be titled and framed more as a review with the meta-analysis being such a small piece. Overall, I would accept this work with minor corrections rearranging and clarifying some of the content.

We agree with reviewer, title was corrected.

Abstract: I think it would be worthwhile to mention an example of SRB. The second to last sentence doesn't really seem to make sense. That point is barely mentioned in the article.

It was deleted.

Body text: 

Line 112: Extra 'o'

It was corrected.

Line 236-241: This seems out of place, perhaps make this a small chart or just a sentence? Why bullet points?

We think that it is ok, because it allows for the reader easier to orient and understand the criteria helped define subgingival bacterial species as periodontal pathogens.

Tables 1 and 2 look nice. 

Thank you!

Would it be possible to move 3.4 to earlier in the text? Same with figure 4, it might benefit the text to put it earlier and include documentation of where those different plaques come from on the diagram. 

It is possible, we have corrected it.

Table 3 has formatting issues.

It was corrected. Thank you!

Pg 10 mentions antibiotics, but this is not discussed in the text in detail. How do antibiotics impact the presence of these microbes? I'd like to see more on that. 

Thank you for your question and interest about antibiotics against SRB, but it should be noted that this group of bacteria is not sufficiently studied under the influence of antibiotics. We have tested some antimicrobial compounds, but for intestinal strains of SRB, not for SRB strains from oral cavity. In the literature, there are no much papers described this topic.

The meta-analysis is really only at the very end of the article and only deals with a few studies. I take it because so little research has been done? The explanation in 5.2 of the graphics (which I think should be a caption) is not quite clear. Are these limitations to these conclusions? How many studies of those examined actually look at SRB? I'm not totally convinced that a forest plot is the best way to address these data, maybe the other reviewers have suggestions.

Yes, it is true. In the literature, there are no much papers describing SRB strains from oral cavity. We have studied recent papers, which are available in the data base of WOS and Scopus and used for meta-analysis. These recent papers are in the references sections. The meta-analysis for SRB from oral cavity has never been conducted before.

Thank you for your time and your help again.

Best Regards,

The Authors